# Decellularized Matrix Induced Spontaneous Odontogenic and Osteogenic Differentiation in Periodontal Cells

**DOI:** 10.3390/biom13010122

**Published:** 2023-01-06

**Authors:** Alexey A. Ivanov, Tamara I. Danilova, Alla V. Kuznetsova, Olga P. Popova, Oleg O. Yanushevich

**Affiliations:** 1Laboratory of Molecular and Cellular Pathology, Department of Paradontology, Evdokimov Moscow State University of Medicine and Dentistry, 20 Delegatskaya Str., 127473 Moscow, Russia; 2Koltzov Institute of Developmental Biology, Russian Academy of Sciences, 26 Vavilov Str., 119334 Moscow, Russia

**Keywords:** decellularized tooth matrix, decellularized periodontal ligament, collagen I hydrogel, osteogenesis, odontogenesis

## Abstract

The regeneration of periodontal tissues is a decisive factor in the treatment of periodontitis. Currently, to achieve complete periodontal regeneration, many studies have evaluated the effectiveness of decellularized tissue-engineered constructs on periodontal regeneration. We studied the possibilities of osteogenic and odontogenic differentiation of periodontal progenitor and stem cells (SCs) of the periosteum and periodontal ligament, in decellularized tooth matrix (dTM) and periodontal ligament (dPDL), in 2D and 3D culture. The cell culture of periodontal cells without decellularized matrices was used as control. On the 14th day of cultivation of PDLSCs, PSCs, and PDLSCs + PSCs on dTM and/or dPDL scaffolds in 2D conditions, in all scaffold variants, a dense monolayer of spindle-shaped cells was intensely stained for markers of osteogenic differentiation, such as osteopontin and osteocalcin. Periodontal cells in the collagen I hydrogel (3D-dimensional culture) were more diverse in shape and, in combination of dTM and dPDL, in addition to osteogenic expression, expressed dentin sialophosphoprotein, an odontogenic differentiation marker. Thus, collagen I hydrogel contributed to the formation of conditions similar to those in vivo, and the combination of dTM with dPDL apparently formed a microenvironment that promoted osteogenic and odontogenic differentiation of periodontal cells.

## 1. Introduction

Periodontal disease is a chronic inflammation of periodontal tissues that leads to progressive destruction of the supporting tissues of the tooth and, ultimately, to tooth loss [1]. The regeneration of periodontal tissues is a decisive factor in the treatment of periodontitis or damaged teeth. Traditional regenerative approaches are aimed at stimulating the growth and differentiation of tissue-resident progenitor cells into fibroblasts, cementoblasts, and osteoblasts, while preventing the germination of epithelial tissues into a periodontal defect. This approach, called guided tissue regeneration, is the use of barrier membranes with or without bioactive molecules, such as enamel matrix derivatives and recombinant growth factors [2]. However, the large heterogeneity of studies confirms the unpredictability of treatments, and none of the existing treatment options provides complete periodontal regeneration [3].

Cellular therapy using stem cells (SCs), such as dental pulp SCs (DPSCs), SCs of apical papilla (SCAP), dental follicle cells (DFCs), cells from the periosteum (PSCs), periodontal ligament stem cells (PDLSCs), and SCs of human exfoliated deciduous teeth (SHED), have been investigated as an alternative approach for their efficacy in periodontal tissue regeneration [4]. In particular, it has been shown that PDLSCs and DPSCs have the ability to differentiate into various cell types with the formation of periodontal ligament, cementum, and alveolar bone [5]. These cells have a better regenerative potential compared to SCs derived from cells of the connective tissue of the gums or the periosteum of the alveolar bone [6].

Numerous approaches to periodontal regeneration using various SCs and scaffolds in the experiment showed encouraging results, but an effective method of periodontal tissue regeneration has not yet been reported [7].

The choice of scaffold is one of the key elements on which the final success in tissue reconstruction largely depends. Scaffolds not only provide cell attachment and growth but also contribute to the formation of the necessary tissue. Commonly used scaffolds, such as ceramics and synthetic or natural polymers, have controlled mechanical properties and good reproducibility, but show low bioactivity [8]. To regenerate periodontal tissues, various methods have been used, involving the application of tissue substitutes, bioactive materials, and synthetic scaffolds. However, all of these treatments have had limited success in the structural and functional regeneration of periodontal tissues [9,10].

To achieve complete periodontal regeneration, many studies have evaluated the effectiveness of decellularized tissue-engineered constructs on periodontal regeneration [11,12,13]. Of particular interest for research in regenerative medicine and dentistry are scaffolds obtained by decellularization of the extracellular matrix (ECM) of mammalian tissues [14]. Such scaffolds do not exhibit immune responses and by their nature contain tissue-specific factors involved in cell growth and differentiation [15,16]. Decellularized ECM (dECM) is an ECM that is devoid of original resident cells and retains the spatial architecture of tissues, is considered a promising natural biomaterial for tissue engineering, designed to support, replace, or repair damaged tissues [14,17,18]. dECMs derived from native tissues such as bone, cartilage, skin, and tooth germs, or from cells such as osteoblasts, chondrocytes, and mesenchymal SCs, have shown promising results in periodontal tissue regeneration [19]. For example, the bone ECM acts as a reservoir of pro-inflammatory cytokines, TGF-β family growth factors, including several BMPs, and angiogenic growth factors such as VEGF, which are required to achieve osteoinduction by regulating various phases of bone regeneration [20,21]. The results of our recent study showed that dental dECM induces spontaneous osteogenic differentiation of periosteal cells in vitro [22].

Naturally, the ECM of periodontal origin can be considered as having an ideal microenvironment (e.g., topography, protein composition) for periodontal regeneration. The proteomic ECM profile of tooth dentin [23,24] and the tooth pulp of healthy human molars [25] and mouse molar periodontal ligament (PDL) [26] have been reported in the past. The ECM of dentin and PDL primarily comprises collagens; collagen type I is the most abundant collagenous protein; collagen types with smaller amounts are type III, type XII, and type V [23,26]. In addition to collagen type I, fibronectin is the second of the predominant proteins present in native PDL [27]. The main noncollagenous proteins in human dentin are dentin sialophosphoprotein (DSPP) and osteopontin (OPN) [23,28].

The use of dECM from a section of a tooth with a periodontal ligament in PDLSC cultivation has shown that this scaffold has a high potential for inducing biomineralization and bone remodeling [12]. Thus, human-derived decellularized tooth matrix (dTM) and/or decellularized PDL (dPDL) can be considered attractive and very promising biomaterials for use in clinical applications for many reasons. Firstly, teeth and PDL are easily collected at any dental clinic. Secondly, no ethical approval is required for tooth collection, because extracted teeth are considered medical waste. Thirdly, their architectonics and macromolecular composition are ideal for the colonization and differentiation of resident SCs from surrounding tissues.

The aim of this study was to investigate the differentiation possibilities of PSCs and PDLSCs on the dECM of the tooth and PDL under 3D versus 2D culture conditions in order to assess the prospects for using cell-free approaches for periodontal tissue repair. We have demonstrated that dECM modulation can direct SC differentiation. Given that various microenvironmental factors mimic cell activity in vivo under 3D conditions, the use of collagen I hydrogel determined odontogenic differentiation of SCs, which was absent in 2D culture.

## 2. Materials and Methods

### 2.1. Preparation of Tooth Crumbs/Particles and Periodontal Ligament Fragments/Strips

Extraction of patients’ healthy teeth (1–3 molars) was carried out in a planned manner for orthodontic reasons. The tissues of the periodontal ligament were gently separated from the surface of the middle third of the root of the patients under aseptic conditions. Strips 0.5–0.7 mm thick were cut with a scalpel and stored in a refrigerator at −20 °C. The teeth were then washed with chlorhexidine. The crown and cementum were removed from the extracted teeth, and particles 1–2 mm in diameter were formed from the remaining tooth roots with pulp using an electric mill (Bosch MKM 6000, Görlingen, Germany) and stored in a refrigerator at −20 °C. Quality control of the formed scaffolds was carried out through a dental microscope (Seiler, St. Louis, MO, USA) at magnification of ×8.

### 2.2. Isolation of Periodontal Cells

SCs and progenitor cells of PDL and periosteum were isolated and cultured according to the procedures previously described [29,30]. Briefly, PDLSCs were isolated from the periodontal ligament of 1–3 molars removed as part of planned orthodontic treatment under sterile conditions. One to two hours after sampling, the biopsy was delivered to the laboratory in a transport medium and subjected to enzyme treatment in a laminar flow cabinet. The tissue was incubated for 70 min at 37 °C in a solution containing 2 mg/mL type I collagenase (Gibco, Grand Island, NY, USA) and 2 mg/mL dispase (Gibco, Grand Island, NY, USA). After enzymatic treatment, the cell suspension was centrifuged twice for 10 min at 600 rpm in culture medium (DMEM, Gibco, USA) and plated in 6-well cell culture plates. Cells were cultured in DMEM-GlutaMAX growth medium (Gibco, USA) supplemented with 15% fetal bovine serum (FBS, Gibco, USA), 100 U/mL penicillin, 100 mg/mL streptomycin and 2 mM essential amino acids (Gibco, USA).

The PSCs were isolated from the periosteal tissue of the alveolar bone 5 × 5 mm in size as part of a planned orthodontic treatment under sterile conditions. The biopsy specimen was delivered to the laboratory in a transport medium 1–2 h after sampling and subjected to enzymatic treatment in a laminar flow hood. After 16–20 h of incubation in type II collagenase solution (Sigma-Aldrich, USA), the crushed tissue homogenate was centrifuged (800 rpm for 5 min at 20 °C), and then the pellet was resuspended in growth medium DMEM-GlutaMAX (Gibco, USA) with the addition of 10% FBS (Gibco, USA) and an antibiotic/antimycotic (Gibco, USA) at standard concentration. The resulting cell suspension was transferred to 25 cm^2^ plastic culture flasks (Corning, Gilbert, AZ, USA) and grown in a CO_2_ incubator at 5% CO_2_ for 10–15 days until a subconfluent monolayer was formed. Upon reaching 90–95% confluent monolayer, the cells were removed with a 0.25% trypsin solution in 1 mM EDTA and subcultured into 25 cm^2^ plastic culture flasks (Corning, Gilbert, AZ, USA) with a density of 1 × 10^5^ cells/mL.

### 2.3. Decellularization

Decellularization was performed according to the technique described previously by sequential incubation in a 1% SDS solution (Sigma-Aldrich, St. Louis, MO, USA), 1% Triton X-100 solution (Sigma-Aldrich, USA) and a DNase solution (Sigma-Aldrich, USA) (20 μg/mL) in 4.2 mM MgCl_2_ (Sigma-Aldrich, USA) [22]. After washing twice with deionized water, the tooth particles and PDL strips were incubated in DMEM culture medium (Gibco, Thermo Fisher Scientific, USA) with antibiotics (300 U/mL penicillin + 300 μg/mL streptomycin + 75 μg/mL amphotericin B) (Gibco, USA). At the end of the decellularization procedure, tooth particles and PDL strips were transferred to fresh DMEM medium (Gibco, USA) with antibiotics and stored at −70 °C. The efficiency of decellularization was validated by staining histological sections with VECTASHIELD Antifade Mounting Medium with DAPI (Vector Laboratories, Inc., Burlingame, CA, USA) under a fluorescence microscope (BX53 Olympus Europa SE & Co., Hamburg, Germany).

### 2.4. Cell Culturing in 2D Conditions

For 2D cultivation, dTM and dPDL were used separately, as well as their composition in a 3:1 ratio. PDLSCs, PSCs, and PDLSCs + PSCs were added to each well in 12-well plates (Corning, USA) at 1.0 × 10^6^ in 1.0 mL of DMEM-GlutaMAX (Gibco, USA) supplemented with 10% FBS (Gibco, USA) and antibiotic/antimycotic (Gibco, USA) at standard concentration. The cells with scaffolds were cultured for 14 days in a CO_2_ incubator at 37 °C. Cell culture of PDLSCs, PSCs, and PDLSCs + PSCs without added scaffolds was used as a control. All experiments were carried out at least twice with three biological replicates each.

### 2.5. Preparation of Collagen I Hydrogel and Bioengineered Constructs

The 3 mg/mL collagen type I stock solution was prepared by diluting freeze dried collagen (10 mg sterile acid-solubilized rat collagen, ~95% collagen I, 5% collagen III; Q C11-NCL, Imtek, Russia) in 0.02 M acetic acid. A working collagen type I solution of 1.2 mg/mL in 10 × Medium 199 (10 × M199, Gibco, USA) was prepared on ice immediately prior to use, neutralizing to pH 7.0 with NaOH and HEPES. This preparation proceeded as follows: (1) 0.34 M NaOH solution was added to Sodium bicarbonate 7.5% solution in a culture tube (tube 1); (2) GlutaMax was added to 10×M199 (tube 2); (3) FBS was added to 1 M HEPES (tube 3); (4)the desired mass of collagen was placed in a separate culture tube (tube 4) and kept on ice; (5) the contents of tubes 1, 2, and 3 were sequentially added to the acidic collagen solution (tube 4) and slowly mixed.

Decellularized matrices, dTM, dPDL, or their combination (3:1), were added to the working collagen type I solution so that the total gel volume increased by less than 25%. Then, a suspension of PDLSCs, PSCs, and PDLSCs + PSCs in a small volume of Medium (~50–100 µL) was added to the working collagen type I solution such that the hydrogel would contain 0.5 × 10^6^ cells/mL.

Neutralized collagen I hydrogel with embedded dTM, dPDL, or their combinations and cell suspension (PDLSCs + PSCs) were placed in the wells of a 12-well plate in a volume of 1.5 mL per well and left to polymerize at 37 °C, 5% CO_2_ for 60 min. After collagen polymerization, prewarmed complete DMEM-GlutaMAX (Gibco) with 10% FBS was added to bioengineered constructs. They were cultured for 14 days in a CO_2_ incubator at 37 °C.

Cell suspension (PDLSCs, PSCs, and PDLSCs + PSCs) in collagen I hydrogel served as a control. All experiments were carried out at least two times with four biological replicates each.

### 2.6. Histological Examination

After 14 days, 2D and 3D cultured cells were fixed with 10% neutral formalin, decalcified, embedded in paraffin, and stained with hematoxylin-eosin according to the standard method.

### 2.7. Immunohistochemical Study

The cultures of PDLSCs and PSCs were incubated with antibodies to CD73 (MA5-15537, Invitrogen, Waltham, MA, USA); CD90 (ab133350, Abcam, Cambridge, UK); STRO-1 (39-8401, Invitrogen). Deparaffinized sections were incubated with antibodies to osteopontin (OPN, ab218237. Abcam), osteocalcin (OC, ab198228, Abcam), and dentin sialophosphoprotein (DSPP, ab216892, Abcam). Prior to immunohistochemical staining, antigen retrieval was performed using Dako PT Link (Dako, Denmark A/S) at 97 °C for 20 min. The unmasking was performed under low pH using EnVison FLEX Target Retrieval Solution (Dako, Glostrup, Denmark A/S). Endogenous peroxidase and host IgG were blocked, and the primary antibodies (dilution 1:200) incubated for 12 h at 4 °C. The EnVision FLEX Detection System (Dako, Denmark A/S) with the chromogen 2, 3-diamino-benzidine DAB (DAB Chromogen Solution, Dako) was used for imaging. Hematoxylin was used to counterstain nuclei. Incubation without primary antibodies was used as a negative control.

### 2.8. Evaluation of Osteogenic and Odontogenic Expression

The number of antigen-positive cells was determined in 6–7 images of random fields of view obtained with a magnification of the microscope lens of 20× by counting in the ImageJ1.48 program (Wayne Rasband, National Institute of Mental Health, Bethesda, MD, USA). The approximate number of total cells contained in the randomly selected field varied from 70 to 1000 for 2D cultures and from 200 to 400 for 3D cultures. The percentage of cells expressing the osteogenic or odontogenic differentiation marker (*M*%) was estimated by dividing the number of positively stained cells (*Np*) by the total number of cells (*Nt*) per field of view in the different groups using the formula:$$M\%=\frac{N_{P}}{Nt}\times 100\%.$$

### 2.9. Statistical Analysis

Statistical processing of results was carried out using the Excel 2016 software (Microsoft, Redmond, WA, USA). Levels of significance were calculated using dispersion analysis (one-way or two-way ANOVA). Probability values of *p* < 0.05 were considered statistically significant. A boxplot was used to plot the minimum, 25th percentile, median (the 50th percentile), means, 75th percentile, maximum, and outlying values.

## 3. Results

### 3.1. Two-Dimensional Cell Culture

On the 14th day of cultivation of PDLSCs, PSCs, and PDLSCs + PSCs on scaffolds of dTM and/or dPDL under 2D conditions, the morphological picture under inverted microscopy was similar and did not depend on the type of scaffold or the type of SCs. A dense monolayer of spindle-shaped cells was formed at both the bottom of the culture wells and around the scaffolds (Figure 1A–C).

Histological examination of the preparations revealed a weak adhesion of PDLSCs, PSCs, and PDLSCs + PSCs to dTM, while when using dPDL, these cells were found both on the surface and inside the decellularized scaffolds (Figure 2A–C).

An immunohistochemical study revealed intense staining of cells for osteogenic differentiation markers, such as OPN and osteocalcin (OC), when using all variants of scaffolds and SCs (Figure 3A–C). Osteogenic differentiation of SCs varied depending on the type of scaffold and was most pronounced on dTM (Figure 3D). Statistical analysis showed that the type of matrix affected the percentage of stained cells, the type of cells did not affect the percentage of stained cells, and there was no cumulative effect (Figure 3D; Appendix A). Immune staining for the DSPP, an odontogenic differentiation marker, was absent when cultured on dTM and dPDL, and their combinations. Immune staining of SCs without scaffolds was negative for osteogenic and odontogenic differentiation markers (Appendix A), and was positive for mesenchymal SC markers (Appendix A).

### 3.2. Three-Dimensional Cell Culture

The culture of PDLSCs, PSCs, and PDLSCs + PSCs with scaffolds in collagen I hydrogel (3D conditions) on day 3 marked the onset of hydrogel contraction (Figure 4A), which increased on day 10 (Figure 4B).

A histological study of collagen hydrogel preparations revealed a similar morphological picture: cultured cells had more pronounced adhesion to dPDL than to dTM (Figure 5A–C). At the same time, in contrast to the 2D culture, the cells in the collagen I hydrogel had a more diverse shape: elongated, spindle-shaped, and rounded.

Immunohistochemical staining revealed osteogenic differentiation of numerous cells, both separately using dTM and dPDL, and their combination (Figure 6A–C). Statistical analysis showed that the type of matrix affected the percentage of stained cells, the type of cells did not affect the percentage of stained cells, and there was no cumulative effect (Figure 6D; Appendix A).

When using a combination of dTM and dPDL, in addition to the cells positive for the osteogenic markers OPN and OC, elongated cells expressing DSPP were found in PDLSCs, PSCs and PDLSCs + PSCs (Figure 7). Statistical analysis showed that the type of cells did not affect the percentage of stained cells (Appendix A). Immune staining of SCs cultured in collagen I hydrogel without scaffolds was negative for osteogenic and odontogenic differentiation markers (Appendix A).

## 4. Discussion

Currently, one of the areas of regenerative medicine that is actively developing is endogenous regenerative medicine, the main goal of which is to create a microenvironment that promotes the initiation, recruitment, and differentiation of resident SCs in the area of damage [9,31,32]. One of the most common approaches to microenvironment modulation is the use of various biomaterials, in particular, decellularized matrices [33]. Decellularized matrices contain various matrix-associated growth factors and adhesive molecules that promote the recruitment and differentiation of resident SCs [21]. However, the specific composition of a decellularized scaffold that promotes cell behaviors, tissue regeneration, and angiogenesis is still unclear, and related cellular and molecular mechanisms are also worth studying [14].

We suggest that in conditions of severe periodontitis, the most likely source of endogenous progenitor and SCs may be the periosteum and undamaged periodontal ligaments. In this regard, in this study, to assess the regenerative potential of decellularized matrices (dTM and dPDL) in the restoration of periodontal tissues, periodontal progenitor and SCs (PDLSCs and PSCs) were used as the most likely candidates. The individual potential of the SCs was evaluated when PDLSCs and PSCs were used separately. The conducted studies showed that SCs, regardless of the source of origin, had a similar differentiation potential when using these scaffolds. In this regard, the main attention was paid to the use of a mixture of these cells, which, in our opinion, most fully simulates the processes in vivo.

Under 2D culture conditions with standard culture medium, PDLSCs, PSCs, and PDLSCs + PSCs actively proliferated and formed a dense monolayer, both on the plastic surface and on dTM and/or dPDL. Under these conditions, the cells expressed only osteogenic markers (OPN and OC), i.e., dTM and dPDL formed a microenvironment that promotes cell differentiation in the osteogenic direction. The expression of another osteogenic marker, alkaline phosphatase, and alizarin red staining were absent in all groups. Obviously, the absence of staining is associated with the use of standard medium without inducing factors and a short cultivation time. OC is known to act as a negative regulator, inhibiting premature or inappropriate mineralization [34,35]. In general, decellularization techniques preserve the ability of bone ECM scaffolds to induce osteogenic differentiation of cells in vitro and to promote angiogenesis and cell infiltration in vivo [36,37].

The most pronounced osteogenic differentiation was detected by the use of dTM, which is not surprising as it contains hydroxyapatite. Furthermore, dTM contains signaling molecules and matrix-associated growth factors (TGF-β, BMP-2) required to initiate the formation of new bone tissue that can be effectively integrated into surrounding tissues [20,21]. Evidently, dPDL contributes to the induction of other lines of cell differentiation, in particular odontogenic, which is not expressed under 2D conditions. It is likely that dPDL contains residual basic fibroblast growth factor (bFGF), vascular endothelial growth factor (VEGF), and hepatocyte growth factor (HGF) [27].

It is known that cell morphology and its migration and proliferation rate, as well as gene expression, differ in two-dimensional and three-dimensional cultures [38,39]. For example, cells cultured under 3D conditions have a more diverse morphology, similar to that observed in vivo [40]. Furthermore, the culture of cells with dECM in three-dimensional structures that mimic the architecture of the original tissue can more accurately assess the effect of the biomaterial on the differentiation potential of the populated cells. Under these conditions, cells can receive mechanical and paracrine signals from the local microenvironment, as occurs naturally [41]. Type I collagen is known to provide a biomimetic environment for 3D cell culture [42]; it creates stiffness around cells, which alters their properties [43]. In addition, the spatiotemporal distributions of oxygen and carbon dioxide, nutrients, and waste are formed in 3D cell culture, as well as the customization of other microenvironmental factors that are known to regulate in vivo activity [44].

We studied the influence of the microenvironment formed by dTM and/or dPDL on the differentiation of PDLSCs, PSCs, and PDLSCs + PSCs in the 3D collagen I hydrogel. All variants of decellularized scaffolds contributed, as in the case of two-dimensional cultivation, to the osteogenic differentiation of cells. Statistical analysis showed that the type of matrix affected the percentage of cells stained with OC, while the type of cells did not. At the same time, only the combination of dTM with dPDL promoted, in addition to osteogenic differentiation of cells, the odontogenic differentiation of cells, which was confirmed by staining for DSPP. Obviously, a compound of matrix-associated factors contained in both dTM and dPDL may be important for this effect. Therefore, 3D cultivation on collagen I hydrogel contributed to the formation of conditions close to those in vivo, and the combination of dTM with dPDL apparently formed a microenvironment that promotes both osteogenic and odontogenic differentiation of PDLSCs, PSCs, and PDLSCs + PSCs. It is obvious that dECM can provide a suitable microenvironment for the controlled release of biological signals, including matrix-associated growth factors that contribute to the modeling of physiological processes, including tissue morphogenesis and regeneration [32].

## 5. Conclusions

The culture of PDLSCs, PSCs, and PDLSCs + PSCs with a combination of dTM and dPDL on a 3D collagen I hydrogel revealed spontaneous osteogenic and odontogenic differentiation of periodontal cells. We suggest that it is this combination of decellularized scaffolds/bioengineered construction that is able to form a local microenvironment that promotes both the osteogenic and the odontogenic differentiation of SCs. The results obtained suggest that such a bioengineered construct in a model experiment will contribute to the formation of a microenvironment that initiates the recruitment of resident SCs to the damage zone and their differentiation into cells that form periodontal tissues. Therefore, the dTM/dPDL combination has a high potential to induce the differentiation of resident SCs into odonto/osteoblasts, which makes them good candidates for future therapeutic applications to the endogenous regeneration of dentin and bone defects and the repair of damaged periodontal structures.

## Figures and Tables

**Figure 1 biomolecules-13-00122-f001:**
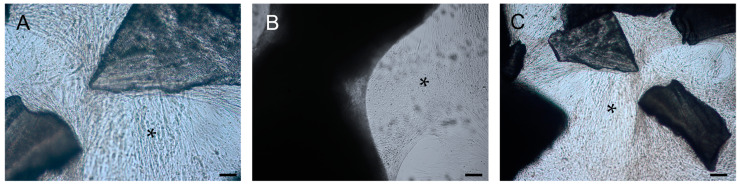
Phase contrast microscopy of decellularized scaffolds cultured with periodontal cells in 2D conditions for 14 days. Scale bars, 200 µm in (**A**–**C**): (**A**) PDLSCs around the dTM; (**B**) PSCs around and in the dPDL; (**C**) PDLSCs + PSCs around dTM and in dPDL. The asterisk indicates monolayer of the cells.

**Figure 2 biomolecules-13-00122-f002:**
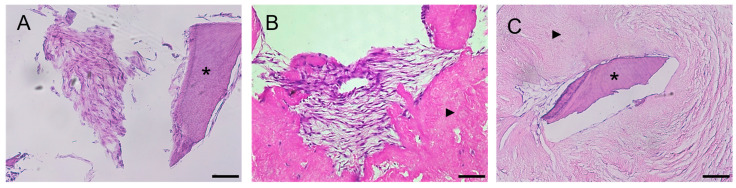
Histological sections of decellularized scaffolds cultured with periodontal cells under 2D conditions for 14 days. Hematoxylin-eosin staining. Scale bars, 100 µm in (**A**), 50 µm in (**B**,**C**): (**A**) PDLSCs next to the dTM indicated by an asterisk; (**B**) PSCs in the dPDL indicated by an arrowhead; (**C**) PDLSCs + PSCs around dTM and in dPDL indicated by an asterisk and arrowheads, respectively; (**A**) original magnification ×200; (**B**,**C**) original magnification ×100.

**Figure 3 biomolecules-13-00122-f003:**
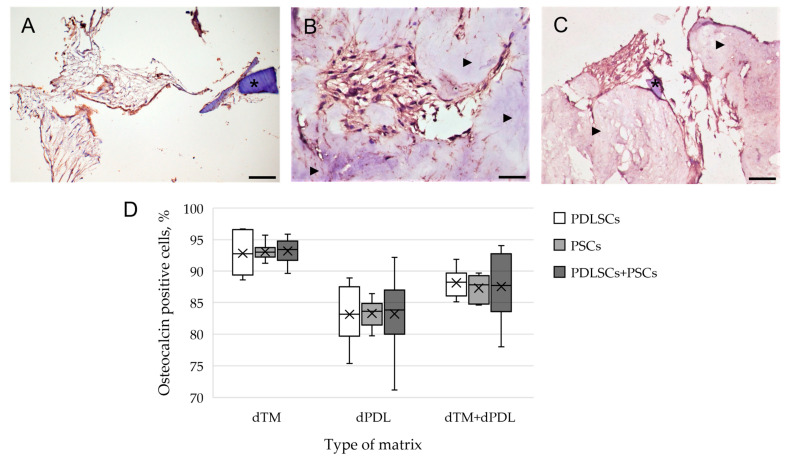
Evaluation of the expression of osteogenic differentiation markers in periodontal cells cultured on decellularized scaffolds under 2D conditions for 14 days: (**A**–**C**) Immunohistochemical staining. The positive cells have a brown color. The nuclei were counterstained with hematoxylin. Scale bars, 100 µm in (**A**–**C**): (**A**) OPN staining of PDLSCs next to the dTM indicated by an asterisk; (**B**) OC staining of PSCs in the dPDL indicated by arrowheads; (**C**) OPN staining of PDLSCs + PSCs around dTM and in dPDL indicated by an asterisk and arrowheads, respectively; (**D**) morphometric analysis of the percentage of cells stained with OC in different groups; *n* = 6. Statistics: two-way ANOVA. *p* < 0.01, significant difference between the groups (the type of matrix).

**Figure 4 biomolecules-13-00122-f004:**
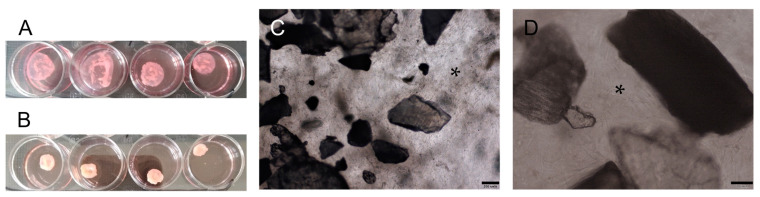
Bioengineered construct, periodontal cells together with scaffolds, in a collagen I hydrogel (3D culture): (**A**) collagen I hydrogel contraction on day 3; (**B**) collagen I hydrogel contraction on day 10; (**C**,**D**) phase contrast microscopy of decellularized scaffolds cultured with PDLSCs + PSCs under 3D conditions for 14 days. Scale bars, 200 µm in (**C**), 100 µm in (**D**). The asterisk indicates cells between decellularized scaffolds.

**Figure 5 biomolecules-13-00122-f005:**
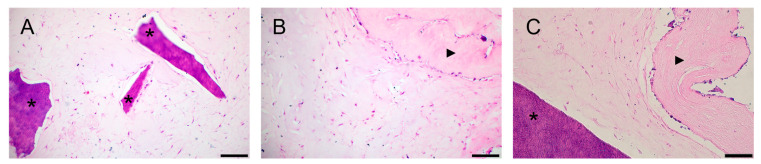
Histological sections of the bioengineered construct, periodontal cells together with scaffolds, in a collagen I hydrogel (3D culture) for 14 days. Hematoxylin-eosin staining. Scale bars, 100 µm in (**A**–**C**): (**A**) PDLSCs next to the dTM indicated by asterisks; (**B**) PSCs in the dPDL indicated by an arrowhead; (**C**) PDLSCs + PSCs next to dTM and around/in dPDL indicated by an asterisk and an arrowhead, respectively.

**Figure 6 biomolecules-13-00122-f006:**
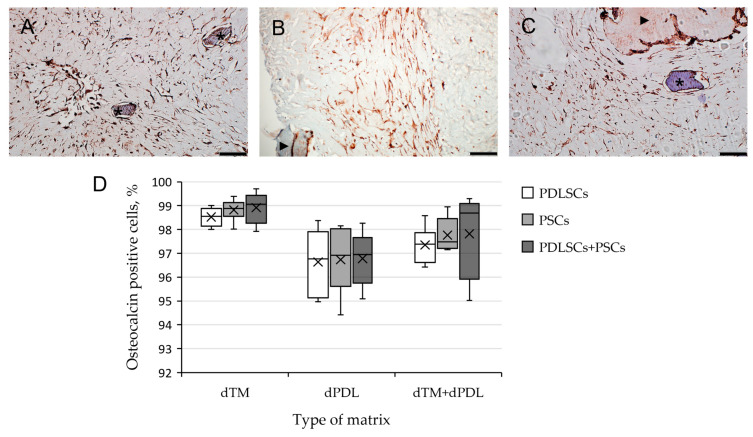
Evaluation of expression of osteogenic differentiation markers in periodontal cells cultured on decellularized scaffolds under 3D conditions for 14 days: (**A**–**C**) Immunohistochemical staining. The positive cells have a brown color. The nuclei were counterstained with hematoxylin. Scale bars, 100 µm in (**A**–**C**): (**A**) OC staining of PDLSCs next to dTM indicated by asterisks; (**B**) OPN staining of PSCs in the dPDL indicated by an arrowhead; (**C**) OC staining of PDLSCs + PSCs around dTM and in dPDL indicated by an asterisk and an arrowhead, respectively; (**D**) morphometric analysis of the percentage of cells stained with OC in different groups; *n* = 6. Statistics: two-way ANOVA. *p* < 0.01, significant difference between the groups (the type of matrix).

**Figure 7 biomolecules-13-00122-f007:**
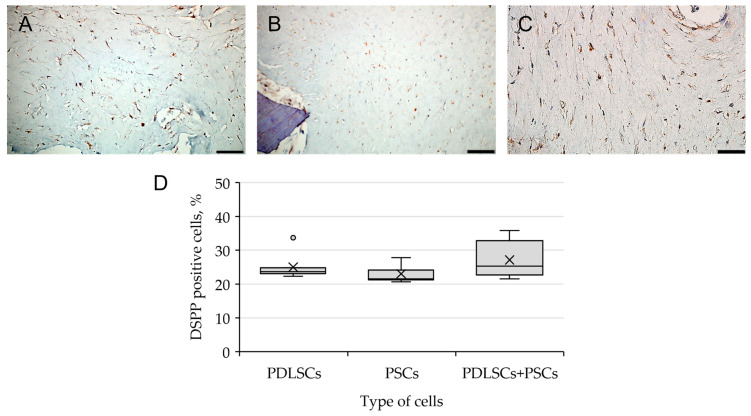
Evaluation of DSPP expression, the marker of odontogenic differentiation, in periodontal cells cultured on dTM + dPDL under 3D conditions: (**A**–**C**) Immunohistochemical staining. The positive cells have a brown color. The nuclei were counterstained with hematoxylin. Scale bars, 50 µm in (**A**–**C**): (**A**) PSCs; (**B**) PDLSCs; (**C**) PDLSCs + PSCs; (**D**) morphometric analysis of the percentage of cells stained with DSPP in different groups; *n* = 7. Statistics: one-way ANOVA. *p* > 0.05, there was no statistically significant difference in DSPP-positive cells between the three groups.

## Data Availability

The data presented in this study are available on request from the corresponding author. The data are not publicly available due to legal issues.

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
