# Peer review of "Decellularized Matrix Induced Spontaneous Odontogenic and Osteogenic Differentiation in Periodontal Cells"

_biomolecules, 2023, doi:10.3390/biom13010122_

Round 1
Reviewer 1 Report
This manuscript presents good quality results the possibilities of differentiation of PSCs and PDLSCs on decellularized matrices of the tooth and periodontal ligament. The authors demonstrate clearly that PDLSCs, PSCs, and PDLSCs + PSCs with a combination of dTM and dPDL on a 3D collagen I hydrogel encouraged spontaneous osteogenic and odontogenic differentiation of periodontal cells. These studies are important for understand of evaluated the effectiveness of decellularized tissue engineered constructs on periodontal regeneration. The paper provides very interesting data but it still needs a several revision.
The following are the comments related:
Introduction
P3.L33: “DTM Particles” What is this? Please add a formal name.
Materials and Methods
P2.L48: “…Planned manner for medical reasons…” Please provide more details.
P2.L49: “…its surrounding tissue was removed…” A more detailed description of the
procedures.
P2.L51: “…leaving fragments of the teeth without enamel…” Have you removed the cementum and pulp? Please elaborate.
P3.L40: Please add the approximate number of total cells contained in the randomly selected field.
Results
Scale bar is lacking. Please add Scale bar to figure.
Figure legends 3D, 6D, 7D Please indicate the number of samples (n= ) used to calculate the mean and SD.
I would suggest that the results be presented in a different way; for example, a table could be used rather than a figure. This would make the results stand out better and make it easierfor the reader to understand the importance them.
Discussion
“...DTM and PDL minimized…” Isn't it a misspelled in "dTM and dPDL" ?
I hope these comments will be helpful.
Author Response
We express our deep gratitude to the reviewer for the hard work of reading the paper and for all the comments and suggestions made for necessary corrections. Below is the list of all the questions and our answers to them.
The following are the comments related:
Introduction
P3.L33: “DTM Particles” What is this? Please add a formal name.
It is just a misspell in Materials and Methods, it is corrected (page 2, line 94; page 3, lines 139,142).
Materials and Methods
P2.L48: “…Planned manner for medical reasons…” Please provide more details.
P2.L49: “…its surrounding tissue was removed…” A more detailed description of the procedures.
P2.L51: “…leaving fragments of the teeth without enamel…” Have you removed the cementum and pulp? Please elaborate.
P3.L40: Please add the approximate number of total cells contained in the randomly selected field.
Agreeing with the reviewer's opinion, we changed the section 'Materials and Methods' (page 2, lines 95-97) and (page 3, lines 99-104). We added the approximate number of total cells contained in the randomly selected field (page5, lines 202-203).
Results
Scale bar is lacking. Please add Scale bar to figure.
Figure legends 3D, 6D, 7D Please indicate the number of samples (n= ) used to calculate the mean and SD.
I would suggest that the results be presented in a different way; for example, a table could be used rather than a figure. This would make the results stand out better and make it easier for the reader to understand the importance them.
In agreement with the reviewer's opinion, we added a scale bar to the figures and the number of samples (n= ) in the figure legends 3D, 6D, 7D. We thought that to compare more than two groups, it would be more appropriate to use analysis of variance (ANOVA) and presented our results in new figures. In these case Standard deviation match Variance. We attached the tables in supplement materials.
Discussion
“...DTM and PDL minimized…” Isn't it a misspelled in "dTM and dPDL" ?
Yes, it is. It is just a misspell, it is corrected (page 13, line 375). Minor grammatical errors and typos have been corrected through the text. All changes to the text are marked in the margin. We hope that this will improve the quality of the manuscript.
Reviewer 2 Report
Comments for the authors
The aim of the present manuscript is to assess the differentiation potential of periodontal dental ligament stem cells (PDLSCs) and periosteum stem cells (PSCs) on decellularized matrices of the tooth and periodontal ligament under 2D and 3D culture conditions. As described in manuscript, decellularized tissue engineered can be used as a scaffold, since theses constructs retain the spatial architecture of tissues, and act as a reservoir of pro-inflammatory cytokines and growth factors, which are required to osteogenic and odontogenic differentiation. Here, the authors tested two types of decellularized tissue engineered: human-derived decellularized tooth matrix (dTM) and decellularized periodontal ligament (dPDL).
Although some interesting results have been shown, some aspects need to be clarified.
-
For dTM preparation, it is not clear if dental cementum was removed or maintained under the samples. This aspect is important since PDL cells only adhere to dentin in the presence of cementum. As shown in figure 2A, PDLSCs are next to dTM, but not adhered to it. This finding is also observed when PDLSCs are cultured in a collagen I hydrogel + dTM (Figure 5A).
-
It is important for the authors to show the images of the negative control group for each antibody, so that we can visualize the specificity of the reaction.
-
How is the behavior of PSCs cultured on dTM? Also, I would like to know if the dTM and dPDL were characterized in relation to their compounds (growth factors, structural proteins and cytokines). This is an important aspect to understand the response of cells. And, this point of view needs to be addressed in the discussion.
-
I would like to hear from the authors why they chose only two markers to assess osteogenic differentiation. Wouldn't it have been more relevant to carry out a proteomics analysis? Further, did it perform a functional assay to evaluate mineral nodule formation, such as alizarin red staining?
Author Response
We express our deep gratitude to the reviewer for the hard work of reading the paper and for all the comments and suggestions made for necessary corrections. Below is the list of all the questions and our answers to them.
Although some interesting results have been shown, some aspects need to be clarified. For dTM preparation, it is not clear if dental cementum was removed or maintained under the samples. This aspect is important since PDL cells only adhere to dentin in the presence of cementum. As shown in figure 2A, PDLSCs are next to dTM, but not adhered to it. This finding is also observed when PDLSCs are cultured in a collagen I hydrogel + dTM (Figure 5A).
Agreeing with the reviewer's opinion, we added in the section 'Materials and Methods' more detailed characteristics for DTM particles (page 3, lines 99-104).
It is important for the authors to show the images of the negative control group for each antibody, so that we can visualize the specificity of the reaction.
We presented the images of the negative control group for each antibody in supplement materials.
How is the behavior of PSCs cultured on dTM?
The behavior of PSCs cultured on dTM was publisher in our article, which presented in reference of our manuscript (21. Ivanov, A.A.; Latyshev, A.V.; Butorina, N.N.; Domoratskaya, E.I.; Danilova, T.I.; Popova, O.P. Osteogenic Potential of Decellularized Tooth Matrix. Bull Exp Biol Med 2020, 169, 512-515; DOI:10.1007/s10517-020-04920-8).
Also, I would like to know if the dTM and dPDL were characterized in relation to their compounds (growth factors, structural proteins and cytokines). This is an important aspect to understand the response of cells. And, this point of view needs to be addressed in the discussion.
In ‘Discussion’ we added the information about matrix-associated factors contained in dTM (page 12, lines 351-353) and dPDL (page 12, lines 355-357).
I would like to hear from the authors why they chose only two markers to assess osteogenic differentiation.
Since the aim of our study was to study the influence of decellularized matrices on differentiation of resident stem cells which may migrated to damage zone for the restoration of periodontal tissues, we choose osteopontin, osteocalcin and dentin sialophosphoprotein (DSPP). Osteopontin is fundamental constituent of cementum matrix and alveolar bone and involved in regulating mineral growth. Osteocalcin is a marker for the maturation of osteoblasts, odontoblasts, and cementoblasts that can regulate the extent of mineralization. DSPP is the classic marker of cementoblasts and cementoblast-like cells (Nanci A, Bosshardt DD. (2006). Structure of periodontal tissues in health and disease. Periodontol 2000 40: 11-28. DOI: 10.1111/j.1600-0757.2005.00141.x)
Wouldn't it have been more relevant to carry out a proteomics analysis?
We are going to continue our study by studying the influence of individual components of the ECM and signaling pathways on periodontal cells differentiation.
Further, did it perform a functional assay to evaluate mineral nodule formation, such as alizarin red staining?
The detection of Ca deposits with alizarin red and detection of alkaline phosphatase commonly used in 14–21-day old cell cultures in an osteo/odontogenic differentiation media. The purpose of our study was to investigate the influence of the local microenvironment of dTM and dPDL on the differentiation of PSCs and PDLSCs. Therefore, we cultured our cells in standard medium (DMEM) for only 14 days. This period in our study is insufficient for the analysis of Ca deposits.
All changes to the text are marked in the margin. We hope that this will improve the quality of the manuscript.
Reviewer 3 Report
Abstract section:
There are ambiguous terms i.e. scaffolds is introduced but the authors do not explain to what exactly they mean by scaffold…since it is a very generic term, widely used in tissue engineering
Introduction section:
There are sentences without an actual meaning, for example “This approach, called guided tissue regeneration, is the use of barrier membranes with or without bioactive molecules, such as enamel matrix derivatives and recombinant growth factor.”….
Results/discussions sections:
Guided tissue regeneration does not rely only on barrier membranes, there are many other approaches that have been addressed within this topic, guided tissue regeneration is a very broad area of study.
Further explanations are needed for explaining these “barrier membranes” . What are they from what materials are they made, what fabrication methods have been used so far?
Why the use of decellularization of mammalian tissues is presented as sECM prototype, but in the text few lines below, the authors state that there are many other dECMs :“dECMs derived from native tissues such as bone, cartilage, skin, and tooth germs, or from cells such as osteoblasts, chondrocytes, and mesenchymal SCs, have shown promising results in periodontal tissue regeneration”…?
The novelty of the study is far from being pointed out, especially given the fact that the authors declare that they already obtained positive results on dECM for periodontal regeneration. What is NEW in the present work? What is the motivation of the study beside what has been already reported in the literature?
In Figure 1 is is not clear where are the cells; the authors should point them by arrows of by other some specific signs in Figure 1. The same observation stands for Figure 4 also.
Much more explanations should be provided for the results from Figure 2.
“At the same time, in contrast to the 2D culture, the cells in the collagen I 251 hydrogel had a more diverse shape: elongated, spindle-shaped, and rounded.”-it is not clear how these cellular shapes can be distinguished in the Figures.
The discussions are weakly supported by appropriate literature (only 4-5 papers are cited as references, the results are not compared to results reported in the literature, so it is not possible to assess the performances of the study).
Th discussions and the conclusions are not expressed in a clear and systematic manner, because of this reason they are difficult to follow.
The whole experiment and the results and discussions are rather superficial and there is not enough novelty of the proposed approach, or at least is is not at all evidenced. The analysis and the interpretation of the results are rather shallow and limited to few general considerations.
For the above reasons, I would recommend the rejection of the manuscript.
Author Response
First of all, we are very grateful to the reviewer for an attentive attitude, great criticism, and certain, critical point of view regarding our paper. Undoubtedly, we take into account all the points of the review. Our answers are briefly given below.
Abstract section:
There are ambiguous terms i.e. scaffolds is introduced but the authors do not explain to what exactly they mean by scaffold…since it is a very generic term, widely used in tissue engineering
When using the term ‘scaffold’, we meant the decellularized tooth matrix and the decellularized periodontal ligament. Therefore, we introduce the term ‘scaffold’ after explaining what we used in our study of decellularized matrices of the tooth and periodontal ligament.
Introduction section:
There are sentences without an actual meaning, for example “This approach, called guided tissue regeneration, is the use of barrier membranes with or without bioactive molecules, such as enamel matrix derivatives and recombinant growth factor.”….
‘This approach, called guided tissue regeneration, is the use of barrier membranes with or without bioactive molecules, such as enamel matrix derivatives and recombinant growth factors [2].’ (page 1, line 37) We have made a link to an article that indicates a fundamental approach in the treatment of periodontitis.
Results/discussions sections:
Guided tissue regeneration does not rely only on barrier membranes, there are many other approaches that have been addressed within this topic, guided tissue regeneration is a very broad area of study. Further explanations are needed for explaining these “barrier membranes”. What are they from what materials are they made, what fabrication methods have been used so far?
It is a text fragment from ‘Introduction’ (page 1, line 37). Guided tissue regeneration (GTR) has been widely used for periodontium regeneration in clinic for decades. It is a regenerative surgical technique that involves the procedure of raising mucogingival flap around affected teeth, scaling and planing root surfaces and placing barrier membranes temporally under gingiva [Gottlow J. New attachment formation as the result of controlled tissue regeneration. J. Clin. Periodontol. 1984;11(8):494–503].
Why the use of decellularization of mammalian tissues is presented as sECM prototype, but in the text few lines below, the authors state that there are many other dECMs :“dECMs derived from native tissues such as bone, cartilage, skin, and tooth germs, or from cells such as osteoblasts, chondrocytes, and mesenchymal SCs, have shown promising results in periodontal tissue regeneration”…?
It is just a misspell in ‘Introduction’, it is corrected (page 2, lines 65-66).
The novelty of the study is far from being pointed out, especially given the fact that the authors declare that they already obtained positive results on dECM for periodontal regeneration. What is NEW in the present work? What is the motivation of the study beside what has been already reported in the literature?
Yes, we received the data about osteogenic differentiation of periostal stem cells on dTM (Ivanov, A.A.; Latyshev, A.V.; Butorina, N.N.; Domoratskaya, E.I.; Danilova, T.I.; Popova, O.P. Osteogenic Potential of Decellularized Tooth Matrix. Bull Exp Biol Med 2020, 169, 512-515; DOI:10.1007/s10517-020-04920-8). In the manuscript submitted, dPDL, like dTM, was shown to induce spontaneous osteogenic differentiation with the use of standard growth medium. Furthermore, we have shown for the first time that a combination of these dECMs under 3D conditions is capable of inducing odontogenic differentiation using standard growth medium.
In Figure 1 is is not clear where are the cells; the authors should point them by arrows of by other some specific signs in Figure 1. The same observation stands for Figure 4 also.
We marked the monolayer of cells with an asterisk in Figures 1 and 4.
Much more explanations should be provided for the results from Figure 2.
We changed the legend to Figure 2.
“At the same time, in contrast to the 2D culture, the cells in the collagen I 251 hydrogel had a more diverse shape: elongated, spindle-shaped, and rounded.”-it is not clear how these cellular shapes can be distinguished in the Figures.
We believe that hematoxylin-eosin and immunohistochemical staining in the Figures makes it possible to determine the shape of the cells.
The discussions are weakly supported by appropriate literature (only 4-5 papers are cited as references, the results are not compared to results reported in the literature, so it is not possible to assess the performances of the study).
We expanded the discussion and added literary references (page 12, lines 347-349; 351-353; 355-357; 367-369). All changes to the text are marked in the margin.
The whole experiment and the results and discussions are rather superficial and there is not enough novelty of the proposed approach, or at least is is not at all evidenced. The analysis and the interpretation of the results are rather shallow and limited to few general considerations. For the above reasons, I would recommend the rejection of the manuscript.
The authors thank the reviewer again. The review significantly helped to improve the text of the article.
Reviewer 4 Report
1) Please add size bars in each figure.
2) Please write in conclusion; how decellularized scaffolds/bioengineered construction help to make an anabolic microenvironment for different types of cells.
Author Response
We thank the reviewer for his work of reading the paper and for all the comments and suggestions made for necessary corrections. Below is the list of all the questions and our answers to them.
1) Please add size bars in each figure.
We added scale bar to figures.
2) Please write in conclusion; how decellularized scaffolds/bioengineered construction help to make an anabolic microenvironment for different types of cells.
In the ‘Discussion’ we added the role of the anabolic microenvironment of 3D culture for different types of cells (page 13, lines 382-385).
We hope that this will improve the quality of the manuscript. All changes to the text are marked in the margin.
Round 2
Reviewer 3 Report
I regret, but the answers to my questions are very difficult to identify in the revision. I suppose that the authors colored the paragraph inserted in the revised manuscript in yellow?
I asked for detailed argumentation of the novelty of the work. I do not find it in the revised manuscript.
The "broad discussion" of the results cannot be found in the revised manuscript, there are around only 7 lines added...
I still do not consider the manuscript as suitable for publication, Biomolecules is a high level Journal-O2 area-.
I leave up to the Editor the final decision, therefore I will not send these comments to the authors.
Author Response
Thank you very much for your opinion.
I regret, but the answers to my questions are very difficult to identify in the revision. I suppose that the authors colored the paragraph inserted in the revised manuscript in yellow?
We made our revisions to manuscript using the “Track Changes” function because it is the requirement of the editorial board.
I asked for detailed argumentation of the novelty of the work. I do not find it in the revised manuscript.
We regret that you did not see the novelty of our work. We believe that we have been able to show that the modulation of the extracellular matrix can direct differentiation of resident stem cells. Furthermore, in 3D cell culture, the spatiotemporal distributions of different microenvironmental factors regulate activities in vivo. This allowed us to detect odontogenic differentiation of stem cells, which was absent in the 2D culture. We hope that these data will contribute to the wider use of cell-free approaches in regenerative medicine.
The "broad discussion" of the results cannot be found in the revised manuscript, there are around only 7 lines added...
Yes, of course, we could greatly expand the 'discussion' section. But we believe that the use of a previously published text (Ivanov, A.A.; Kuznetsova,A.V.; Popova, O.P.; Danilova, T.I.; Yanushevich, O.O. Modern Approaches to Acellular Therapy in Bone and Dental Regeneration. Int. J.Mol. Sci. 2021, 22, 13454. https://doi.org/10.3390/ijms222413454) violates the laws of scientific ethics. Therefore, we added only those data, which required additional discussion.
I still do not consider the manuscript as suitable for publication, Biomolecules is a high level Journal-O2 area-. I leave up to the Editor the final decision, therefore I will not send these comments to the authors.
The authors thank the reviewer again. The review significantly helped to improve the text of the article.